# Comprehensive Analysis of Gene Expression Profiling to Explore Predictive Markers for Eradication Therapy Efficacy against *Helicobacter pylori*-Negative Gastric MALT Lymphoma

**DOI:** 10.3390/cancers15041206

**Published:** 2023-02-14

**Authors:** Hidehiko Takigawa, Ryo Yuge, Ryo Miyamoto, Rina Otani, Hiroki Kadota, Yuichi Hiyama, Ryohei Hayashi, Yuji Urabe, Kazuhiro Sentani, Naohide Oue, Yasuhiko Kitadai, Shiro Oka, Shinji Tanaka

**Affiliations:** 1Department of Endoscopy, Hiroshima University Hospital, Hiroshima 734-8551, Japan; 2Department of Gastroenterology, Hiroshima University, Hiroshima 734-8551, Japan; 3Department of Clinical Research Center, Hiroshima University Hospital, Hiroshima 734-8551, Japan; 4Gastrointestinal Endoscopy and Medicine, Hiroshima University Hospital, Hiroshima 734-8551, Japan; 5Department of Molecular Pathology, Hiroshima University Hospital, Hiroshima 734-8551, Japan; 6Department of Health and Science, Prefectural University of Hiroshima, Hiroshima 734-8558, Japan

**Keywords:** gastric mucosa-associated lymphoid tissue lymphoma, non-*Helicobacter pylori* helicobacters, *Helicobacter pylori*, predictive marker, RNA sequencing, comprehensive analysis, eradication therapy

## Abstract

**Simple Summary:**

In *API2-MALT1*-positive gastric mucosa-associated lymphoid tissue (MALT) lymphoma, eradication therapy is known to be ineffective. Among *API2-MALT1*-negative cases, eradication therapy is effective for *Helicobacter pylori* (Hp)-positive cases and partially effective for Hp-negative cases. Herein, we explored predictive markers for eradication therapy efficacy in cases that were negative for both *API2-MALT1* and Hp. Among 164 gastric MALT lymphoma patients, 36 were negative for both *API2-MALT1* and Hp. We divided cases of MALT lymphoma negative for both *API2-MALT1* and Hp into complete-response (CR) and no-change (NC) groups based on eradication therapy efficacy and conducted comprehensive gene expression analysis. Pathway analysis showed that cancer- and infection-related genes were highly expressed in the NC and CR groups, respectively. Sixteen candidate genes for predictive markers were extracted and validated with real-time PCR. Olfactomedin-4 and Nanog homeobox were positive and negative predictive factors, respectively, for eradication therapy efficacy against gastric MALT lymphoma; they were negative for both *API2-MALT1* and Hp.

**Abstract:**

Although radiotherapy is the standard treatment for *Helicobacter pylori* (Hp)-negative gastric mucosa-associated lymphoid tissue (MALT) lymphoma, eradication therapy using antibiotics and an acid secretion suppressor can sometimes induce complete remission. We explored predictive markers for the response to eradication therapy for gastric MALT lymphoma that were negative for both *API2-MALT1* and Hp infection using comprehensive RNA sequence analysis. Among 164 gastric MALT lymphoma patients who underwent eradication therapy as primary treatment, 36 were negative for both the *API2-MALT1* fusion gene and Hp infection. Based on eradication therapy efficacy, two groups were established: complete response (CR) and no change (NC). The Kyoto Encyclopedia of Genes and Genomes pathway analysis showed that cancer-related genes and infection-related genes were highly expressed in the NC and CR groups, respectively. Based on this finding and transcription factor, gene ontology enrichment, and protein–protein interaction analyses, we selected 16 candidate genes for predicting eradication therapy efficacy. Real-time PCR validation in 36 Hp-negative patients showed significantly higher expression of olfactomedin-4 (*OLFM4*) and the Nanog homeobox (*NANOG*) in the CR and NC groups, respectively. *OLFM4* and *NANOG* could be positive and negative predictive markers, respectively, for eradication therapy efficacy against gastric MALT lymphoma that is negative for both *API2-MALT1* and Hp infection.

## 1. Introduction

*Helicobacter pylori* (Hp) infections and *API2-MALT1* fusion gene are etiologic factors of gastric mucosa-associated lymphoid tissue (MALT) lymphoma [1]. However, 23% of gastric MALT lymphoma patients are Hp-negative, and 93% do not harbor the API2-MALT1 translocation in stage I gastric MALT lymphoma [2]. Recently, autoimmune diseases [3,4,5] and some infections caused by bacteria other than Hp [6,7] have been identified as pathogens of these double-negative gastric MALT lymphomas. Non-*Helicobacter pylori* helicobacter (NHPH) is involved in the pathogenesis of gastric MALT lymphoma [8,9]. The PCR diagnostic method for detecting five NHPH species (*H. suis*, *H. bizzozeronii*, *H. felis*, *H. salomonis*, and *H. heilmannii s.s.*) [8], the DNA extraction method from formalin-fixed paraffin-embedded (FFPE) tissues [10,11], and the culture method for *H. suis*, known as the major species among NHPHs [12], have been established. Thus, the number of reports describing the relationship between NHPH infection and gastric MALT lymphoma has been increasing [13,14,15,16].

In Hp-positive gastric MALT lymphoma, eradication therapy has a high response rate of 77–87% [1,9,17]; hence, the National Comprehensive Cancer Network guidelines recommend eradication therapy as the initial treatment [18]. However, in Hp-negative gastric MALT lymphoma, radiation therapy (RT) and chemotherapy are recommended, owing to the low response to eradication therapy [18]; however, a complete response of MALT lymphoma to eradication therapy has recently been reported, even in Hp-negative cases [19,20]. Considering the invasiveness of RT, it has been argued that eradication therapy should be attempted even in Hp-negative cases [21].

Several studies have reported predictive factors for the response to eradication therapy in gastric MALT lymphoma. The *API2-MALT1* fusion gene is a well-known predictor of resistance to eradication therapy in gastric MALT lymphoma [21,22]. Other predictive factors, including nuclear expression of BCL10 and NF-κB [22], NK cell infiltration, expression of CD86 [23], a large proportion of CD19- and CD20-positive cells [24], high expression of cMET [25], high microsatellite instability (MSI) [26], and high values of Hp and CagA antibodies [27], contribute to poor responsiveness to eradication therapy in gastric MALT lymphoma. However, since Hp-negative cases are relatively rare, no reports have revealed predictive factors for eradication therapy focusing on Hp-negative gastric MALT lymphoma using comprehensive analysis methods.

We have previously evaluated NHPH infections in a series of gastric MALT lymphoma patients who were negative for both the *API2-MALT1* fusion gene and Hp infection [9]. In these subjects, eradication therapy was significantly more effective in NHPH-positive cases than in NHPH-negative cases, and NHPH infection was reported as a positive predictive marker for complete response (CR) by eradication therapy. However, 25% of the patients failed to respond to eradication therapy, even though they were NHPH infection-positive. In contrast, 23% of patients achieved CR following eradication therapy, despite testing negative for NHPH infection. These results suggest that factors other than NHPH infection may be involved in the efficacy of eradication therapy in gastric MALT lymphoma cases that are negative for both *API2-MALT1* and Hp infection.

Therefore, we aimed to explore genes that may be predictive factors for the efficacy of eradication therapy in *API2-MALT1*-negative and Hp-negative gastric MALT lymphoma cases, based on a comprehensive analysis using RNA sequencing.

## 2. Materials and Methods

### 2.1. Patients

Patients with gastric MALT lymphoma treated at our center between October 2006 and September 2020 and who received eradication therapy as frontline treatment were included in this study. Patients who received any other frontline treatment and lost follow-up cases were excluded. Treatment outcomes for all 137 cases were examined according to *API2-MALT1* mutation and Hp infection status. Subsequently, in cases negative for both *API2-MALT1* and Hp infection, the impact of the gene expression profile on response to eradication therapy was analyzed.

### 2.2. Evaluation of the API2-MALT1 Chimeric Transcript and Hp Infection

All cases were examined for Hp infection and the presence of the *API2-MALT1* chimeric transcript. Hp infection was endoscopically and serologically evaluated using an anti-Hp IgG antibody (E-Plate EIKEN HpAb, Eiken Chemical Co., Ltd., Tokyo, Japan). Successful eradication was confirmed using a urea breath test. Fluorescence in situ hybridization (FISH) analysis was conducted by LSI Medience Corporation (Tokyo, Japan) on fresh biopsy samples from all patients to detect the *API2-MALT1* chimeric transcript, as previously described [28,29]. 

### 2.3. Therapy Evaluation

Details about eradication therapy are described in Appendix A. Briefly, Hp-positive cases with persistent Hp after frontline eradication therapy were eligible for second-line eradication therapy. Conversely, in Hp-negative gastric MALT lymphoma cases, eradication therapy was conducted only once with the first-line eradication therapy regimen. Following treatment, histopathological evaluation was performed using the Group d’Etude des Lymphomes de l’Adulte (GELA) grading system [30]. The complete response (CR) was defined as a complete histological response (ChR) or probable minimal residual disease (pMRD), whereas no change (NC) was defined as responding residual disease (rRD) or no change (NC). Patient characteristics, treatments, and outcomes were retrospectively analyzed. After eradication therapy, all patients were followed up endoscopically every 4 months for the first year and annually thereafter.

### 2.4. Quantitative Reverse Transcription Polymerase Chain Reaction

Quantitative reverse transcription polymerase chain reaction (qRT-PCR) was performed using a LightCycler FastStart DNA Master SYBR Green I Kit (Roche Diagnostics, Basel, Switzerland). The detailed methodology is presented in the Appendix A. Primer sequences are listed in Appendix A.

### 2.5. RNA Sequencing

Tumor RNA was extracted from the tumor lesions of gastric MALT lymphoma from two cases in the CR group and two cases in the NC group using the RNeasy Mini kit (Qiagen). A series of experiments (total RNA sample quality check, RNA sequencing, sequencing read filtering, and bioinformatics analysis) were conducted by the Beijing Genomics Institute (Beijing, China) as previously reported [31]. Library construction and data processing were performed by the Beijing Genomics Institute (Beijing, China). Concentration was measured using ExKubit dsDNA HS Assay Kits (Shanghai ExCell Biology, Inc., Shanghai, China) and a Fluostar Omega Microplate Reader (BMG Labtech GmbH, Offenburg, Germany). Fragment size was detected using a DNA 1000 Kit (Agilent Technologies, Inc., Santa Clara, CA, USA) and 2100 Bioanalyzer Instrument (Agilent Technologies, Inc.). Libraries were sequenced on a DNBSEQ-G400RS platform, and high-quality reads were aligned to the human reference genome (GRCh38). The sequencing kit was the DNBSEQ-G400RS High-throughput Sequencing Set (FCL PE100) (MGI Tech Co., Ltd., Shenzhen, China), and paired-end sequencing (2 × 100 bp) was performed. Concentration was measured using a Qubit™ ssDNA Assay Kit (Invitrogen; Thermo Fisher Scientific, Inc., Waltham, MA, USA) and Qubit 4 Fluorometer (Invitrogen; Thermo Fisher Scientific, Inc.). The loading concentration was 8–20 ng/µL. The genome reference was GCF_000001405.38_GRCh38.p12. The software used to analyze the data was as follows: Filter: SOAPnuke-1.5.6, Alignment hisat: Hisat2-2.1.0 [32], Alignment bowtie: Bowtie2-2.3.4.3 [33], Expression RSEM: rsem_calculate_expression rsem-1.2.28-0 [34], SNP INDEL: GenomeAnalysisTK [35], Structure Fusion ericscript: Ericscript, Structure AS rMATS: rMATS.3.2.5 [36]. We detected differentially expressed genes (DEGs) with DEseq2, as described previously [37] (Parameters: Fold Change ≥ 2.00 and Adjusted *p*-value ≤ 0.05). Gene Ontology (GO; geneontology.org) and Kyoto Encyclopedia of Genes and Genomes (KEGG; http://www.genome.jp/kegg/, accessed on 1 January 2022) pathway analyses were performed using the RNA Data Visualization System Dr. TOM (Beijing Genomics Institute; http://www.bgi.com/global/dr-tom/, accessed on 1 January 2022), a BGI in-house customized data mining system that combines different published software. 

### 2.6. Analysis Flow

To explore the predictive factors for the efficacy of eradication therapy in Hp-negative gastric MALT lymphoma, we separated the 36 *API2-MALT1* and Hp-negative gastric MALT lymphoma cases (Figure 1) into two groups: 17 cases in which CR was achieved through eradication therapy were defined as the CR group, and 19 cases in which eradication therapy was not effective were defined as the no-change (NC) group. Subsequently, we randomly selected two patients from each group (CR and NC groups) and extracted RNA from tumor tissues and conducted a comprehensive RNA sequencing analysis. The genes highly expressed in the CR group and those highly expressed in the NC group were selected by comprehensive RNA sequencing analysis, and bioinformatics analysis (KEGG, PPI, and transcription factor analysis) was performed. Candidate genes that characterized the CR and NC groups and could be predictors of treatment efficacy were selected, and validation of these candidate genes was conducted using real-time PCR in tumor samples from the 17 CR and 19 NC patients (Figure 2A).

### 2.7. Statistical Analysis

Between-group differences were evaluated using the Mann–Whitney *U* test for quantitative data and χ^2^ test for categorical data. Fisher’s exact test was performed as required. All tests were two-sided, and a *p*-value < 0.05 was considered statistically significant. All analyses were performed using EZR (Saitama Medical Centre, Jichi Medical University, Saitama, Japan) [38]. The significance of DEGs in RNA sequences and mRNA expression in the validation study by real-time PCR was expressed as a q-value, representing a false discovery rate (FDR)-adjusted *p* < 0.05. The cut-off value in the receiver operating characteristic (ROC) curve was calculated based on the Youden index.

Details of other experimental procedures are available in the Appendix A.

## 3. Results

### 3.1. Patients

A total of 164 patients with localized (Lugano stages I and II1) gastric MALT lymphoma were identified, and 27 patients were excluded because they received other treatment or no follow-up was taken (radiation therapy: 14; chemotherapy: 3; ESD: 1; no treatment: 3; lost follow-up: 6). 137 patients were enrolled in this study (Figure 1). The backgrounds and clinicopathological characteristics of Hp-positive and Hp-negative cases are listed in Appendix A. There were no significant differences in age, sex, number of lesions (multiple or single), location, or morphological type between the Hp-positive and Hp-negative cases. Hp-positive patients showed more mucosal atrophy and responded significantly better to eradication therapy than Hp-negative patients (*p* < 0.001). Among 87 cases of Hp-positive patients, 74 cases were successfully eradicated by first-line eradication therapy, ten cases underwent second-line eradication therapy, and Hp infection was successfully eradicated in five of these cases.

*API2-MALT1*, *API2-MALT1* chimeric transcript; chemo, chemotherapy; Hp, *Helicobacter pylori*; CR, complete response (ChR and pMRD according to the GELA histological grading system); NC, no change (rRD and NC according to the GELA histological grading system); RT, radiation therapy

### 3.2. Treatment Outcomes by Eradication Therapy According to API2-MALT1 Chimeric Transcript Status and Hp Infection Status

In 137 evaluable patients, we first examined the expression of the *API2-MALT1* chimeric transcript in 137 patients who underwent Hp eradication therapy as first-line treatment. Seventeen patients (17/137; 12.4%) were positive for *API2-MALT1*. Hp eradication therapy was not effective in any of these cases, regardless of the Hp infection status. RT, or chemotherapy, was selected as second-line therapy, and CR was achieved in all cases. Among the 120 *API2-MALT1* (−) patients, 84 were infected with Hp. The CR rates after eradication therapy were 76% (64/84) and 47% (17/36) in the Hp (+) and Hp (−) groups, respectively. Eradication therapy was effective in half of the Hp (−) cases (Appendix A and Figure 1). Thus, focusing on 36 cases in which both Hp and *API2-MALT1* chimeric transcript statuses were negative, we analyzed the differences between CR and NC cases. 

### 3.3. Comparison between the CR and NC Groups in Gastric MALT Lymphomas Negative for Both Hp and the API2-MALT1 Chimeric Transcript

Table 1 shows the results of comparisons between the CR and NC groups. Patients in the CR group tended to be younger than those in the NC group. There were no differences in sex, degree of atrophy, number of lesions, localization of lesions, morphological type, or background differences between the two groups. 

### 3.4. DEGs in Comparison between CR and NC Groups in RNA Sequences

According to the analysis flow (Figure 2A), we randomly selected two cases from the CR and NC groups and conducted RNA sequencing using RNA extracted from the six cases. Seventeen genes were more than twice as highly expressed in the CR group, and 85 genes were more than twice as highly expressed in the NC group (Figure 2B,C). 

### 3.5. Transcription Factor Prediction of DEGs

Using RNA sequence data, we predicted DEGs with the ability to encode transcription factors. Simultaneously, we classified the family of transcription factors to which the DEGs belonged. The expression levels of transcription factors in each sample were clustered, and the results are shown in Figure 3A. The transcription factors corresponding to the DEGs are shown in Figure 3B. A list of DEGs with the ability to encode transcription factors in this project is shown in Figure 3C. Zinc finger protein 556 (ZNF556; reference sequence database number: NM_0013000843), belonging to the zf-C2H2 family, was mostly upregulated in the NC group. 

### 3.6. Protein–Protein Interaction Networks of DEGs

We used the STRING (https://string-db.org/, accessed on 1 April 2020) [39] database to analyze PPIs and construct DEG interaction networks. Gene ID 58, actin alpha 1, and skeletal muscle (ACTA1) had the highest PPI score (Figure 3D). The top 100 interaction networks are shown in Figure 3E.

### 3.7. Pathway Analysis of DEGs

The top 20 pathways with large expression changes between the CR and NC groups according to KEGG pathway analysis were extracted. Cancer- and infection-related pathways differed in expression between the NC and CR groups, respectively (Figure 3F, bubble chart). The percentages of upregulated genes in the CR and NC groups were evaluated (Figure 3F, bar chart). The expression of genes involved in infection-related pathways was upregulated in the CR group. The expression of genes related to cancer-related pathways was upregulated in the NC group (Figure 3F, bubble chart). 

### 3.8. Predictive Markers for Eradication Therapy against Hp-Negative Gastric MALT Lymphoma

Based on the finding that genes related to infection- and cancer-related pathways were upregulated in the CR and NC groups, respectively, in the KEGG pathway analysis, infection- and cancer-related genes among the DEGs upregulated in the CR and NC groups, respectively, were selected as candidate genes for positive and negative predictive markers for the efficacy of eradication therapy. Furthermore, based on transcription factor analysis and PPI analysis, more candidate genes for predictive markers were selected among DEGs and added to the list of candidate genes. In total, seven genes upregulated in the CR group and nine genes upregulated in the NC group were selected as candidate genes for predictive markers. Table 2 lists the expression profiles of the candidate genes obtained by RNA sequencing. All 16 candidate genes had large fold changes and a significant FDR (adjusted *p*-value) that met the sufficient significance level.

### 3.9. Real-Time PCR Validation

The expression levels of each candidate gene were analyzed using qPCR in 17 patients in the CR group and 19 patients in the NC group as the validation study (Table 3). Among the seven candidate genes highly expressed in the CR group, nuclear pore complex interacting protein family member B3 (*NPIB3*) and olfactomedin-4 (*OLFM4*) were significantly highly expressed in the CR group by real-time PCR. Similarly, among the nine candidate genes highly expressed in the NC group, Nanog homeobox (*NANOG*) and *ZNF556* were significantly highly expressed in the NC group. To correct for multiple comparisons, we performed an FDR correction for each P-value. According to the correction, *OLFM4* was significantly highly expressed in the CR group, and *NANOG* was significantly highly expressed in the NC group. *OLFM4* and *NANOG* were identified as predictive factors for the efficacy of eradication therapy.

### 3.10. Predictability of NANOG and OLFM4

The NANOG expression was lower in the NC than in the CR groups (Figure 4A), while the OLFM4 expression was higher in the CR than in the NC groups (Figure 4B). According to the ROC curve (Figure 4C,D), NANOG is a negative predictive marker for CR; in cases where NANOG is <3.486, the sensitivity and specificity for CR prediction are 82% and 79%, respectively. Conversely, OLFM4 is a positive predictive marker for CR; in cases where OLFM4 is >0.500, the sensitivity and specificity for CR prediction are 59% and 84%, respectively (Figure 4E).

## 4. Discussion

Here, we explored the predictive factors for the response to eradication therapy in Hp-negative gastric MALT lymphoma. Approximately half (47%) of the *API2-MALT1*-negative cases of Hp-negative gastric MALT lymphoma responded to eradication and achieved CR. To identify predictive factors for response to eradication therapy in these cases, gene expression profiling was comprehensively investigated via RNA sequencing, and *NANOG* and *OLFM4* were identified as candidates for predictive markers for response to eradication therapy. 

Eradication was ineffective in the API2MALT1-positive patients, with a success rate of 76% in Hp-positive API2MALT1-negative cases and 47% in Hp-negative cases. The success rate of eradication in the Hp-negative cases was higher than that previously reported (15.5–28%) [21] [40]. This difference may be because the previous studies did not account for *API2MALT1* in evaluating the CR rate in Hp-negative gastric MALT lymphoma.

KEGG pathway analysis based on RNA sequencing results showed that infection-related genes were upregulated in the CR group, while cancer-related pathways were upregulated in the NC group. These results may explain why the CR group responded well to eradication therapy using antibiotics, whereas the NC group did not. Consistent with our previous study showing that Hp-negative gastric MALT lymphoma, which is considered to be caused by NHPH infection, responds well to eradication therapy [9], the results suggest that some types of bacterial infection, including NHPH, may be involved in the pathogenesis of Hp-negative gastric MALT lymphoma that responds to eradication therapy with antibiotics.

Focusing on the infection-related genes and cancer-related genes, we identified 16 candidate genes among the DEGs as predictive factors for the efficacy of eradication therapy in Hp-negative gastric MALT lymphoma based on bioinformatics analysis. Subsequently, we conducted real-time PCR validation in all Hp-negative gastric MALT lymphoma patients and found that *OLFM4* was significantly upregulated in the CR group, and *NANOG* was significantly upregulated in the NC group, suggesting that these genes may be predictive factors for the efficacy of eradication therapy in Hp-negative gastric MALT lymphoma. 

NANOG, a crucial transcription factor in embryogenesis and tumorigenesis, is overexpressed in most cancer stem cells (CSCs) [41,42]. NANOG is found in embryonic stem cells and CSCs and plays a central role in maintaining the self-renewal and pluripotency capacities of stem cells [43]. High NANOG expression is correlated with tumor progression and poor differentiation in various cancers [44]. CSCs inhibit antitumor immunity, are associated with low immunogenicity and immunosuppression, protect tumors from an antitumor immune response, and adjust to the unfavorable tumor microenvironment conditions caused by chemotherapy [45]. Comprehensive microarray analysis has revealed that diffuse large B-cell lymphoma (DLBCL) with higher expression of CSC markers, such as NANOG, is highly progressive [46]. Considering the above characteristics of NANOG, which are associated with the stemness and aggressiveness of cancer, the upregulation of *NANOG* in the NC group suggests that *NANOG* may serve as a negative predictive marker for the efficacy of antibiotic therapy in eradication therapy.

In this study, OLFM4 was extracted as a positive predictive marker for the efficacy of eradication therapy in Hp-negative gastric MALT lymphoma. OLFM4 functions in intestinal stem cells and CSCs [47] and is related to cancer progression as a cancer-promoting factor [48,49], the features of which are partially similar to those of NANOG.

However, its function varies and has been reported as a glycoprotein negatively regulating the host defense system against bacterial infection [50]. OLFM4 expression is significantly upregulated in the intestinal epithelium in inflammatory bowel disease [51,52]. Microarray analysis has shown that OLFM4 expression is significantly upregulated in the gastric mucosa of Hp-infected patients compared with that in uninfected controls [53]. OLFM4 overexpression caused by Hp infection induces the activation of neutrophil and macrophage infiltration and direct action on epithelial cells, indicating a potential role for OLFM4 in the host immune response against *H. pylori* infection [54]. The upregulation of *OLFM4* expression in the CR group suggests that environmental conditions at the tumor site, including bacterial infections other than Hp and the associated inflammation, may contribute to the response to eradication therapy. *OLFM4* may serve as a positive predictive marker for the efficacy of antibiotic therapy in eradication therapy. In the present context, where eradication therapy is widespread, the number of Hp-negative gastric MALT lymphoma cases is expected to increase. From these Hp-negative cases, identifying those cases that respond to eradication therapy will be clinically important for decision-making regarding treatment. Additionally, the fact that RT can be avoided if eradication therapy is successful is important from a medical economics perspective. Although NHPH infection status is considered one of the major factors involved in the response to eradication therapy, there is currently no established method for the diagnosis of NHPH infection. The gold-standard PCR method is not easy to use in clinical settings and is not an absolute method owing to the possibility of false negatives, because it uses biopsy tissue samples, which contain only tiny amounts of bacterial DNA. 

However, the sensitivity and specificity of NANOG and OLFM4 calculated in this study (NANOG:; sensitivity 82%, specificity 79%, OLFM4:; sensitivity 59%, specificity 84%) are insufficient for application in clinical practice. Our study generated only a hypothesis, though it might give insights for further studies to explore the predictive markers and clarify the pathogen of gastric MALT lymphoma. The application of the two genes identified here as predictive markers in clinical practice should be assessed in the future. If the expression of proteins coded by *NANOG* and *OLFM4* is evaluated via immunostaining using biopsy tissue samples, the localization of gene expression will be clarified, and these genes can then be used as simple indicators. 

This study has some limitations. First, the number of patients for RNA sequence analysis might not be adequate. However, we believe that the validation study performed after RNA sequences were analyzed included an adequate number of patients to evaluate statistical significance. Second, the presence or absence of NHPH, one of the important pathogens of gastric MALT lymphoma, was not considered in this study. Nevertheless, as there are no reports focusing on the response to treatment of Hp-negative gastric MALT lymphoma, the results of this study are meaningful. Third, eradication regimens were not strictly unified in all cases; as this was a retrospective study, some bias may exist. Lastly, the evaluation of gene mutations by next-generation sequences (NGS) could not be conducted in this study. Integrating the data on the gene expression profiles obtained in this study with the mutational analyses using NGS could make the prediction more precise and accurate. The limitations of this study point to some future challenges in the field that need to be solved.

Overall, in this study, we explored predictive markers for the efficacy of eradication therapy against gastric MALT lymphoma that was negative for both the *API2-MALT1* fusion gene and Hp infection. Low expression of cancer-related genes, such as *NANOG*, and high expression of infection-related genes, such as *OLFM4*, could be predictive markers for the efficacy of eradication therapy. Identification of predictive factors is important in decision-making regarding treatment strategies for gastric MALT lymphoma and may aid the decision of whether eradication therapy should be used or whether other treatment options, such as RT, should be selected. 

## 5. Conclusions

We determined predictive markers for eradication therapy in gastric MALT lymphoma that were negative for both *API2-MALT1* and Hp. Further validation is required by evaluating the protein expression of these predictive markers via immunohistochemistry or FISH using tumor tissues and biopsy samples.

## Figures and Tables

**Figure 1 cancers-15-01206-f001:**
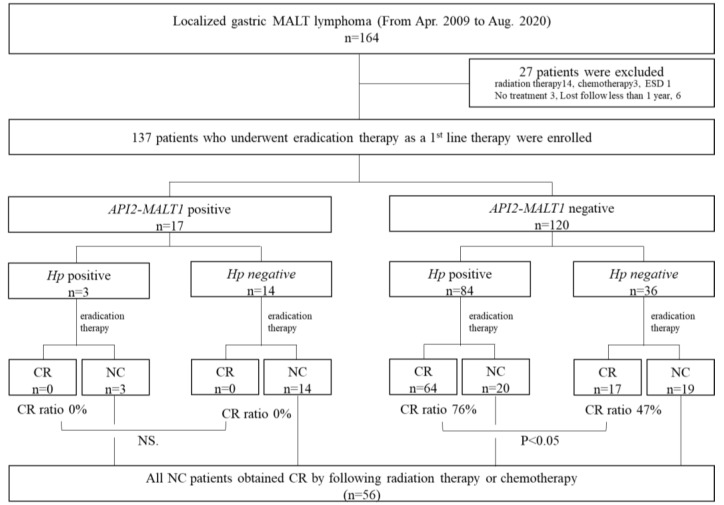
Inclusion and exclusion criteria and treatment outcomes. From 2009 to 2020, 164 patients were diagnosed with localized gastric MALT lymphoma in our hospital. Twenty-seven cases who underwent other treatments (radiation therapy, chemotherapy, ESD, and no treatment) as first-line therapy and who could not be followed up for more than 1 year were excluded from this study. Flowchart of treatment administered to patients classified by the *API2-MALT1* chimeric transcript and Hp infection status. All patients underwent eradication therapy as first-line therapy. Non-responders to eradication therapy achieved a CR after RT. There were 17 *API2-MALT1* (+) patients. None of these patients achieved CR after eradication therapy; however, all achieved CR after RT. Among the *API2-MALT1* (−) cases, 76% of the Hp (+) cases and 47% of the Hp (−) cases achieved CR after eradication therapy.

**Figure 2 cancers-15-01206-f002:**
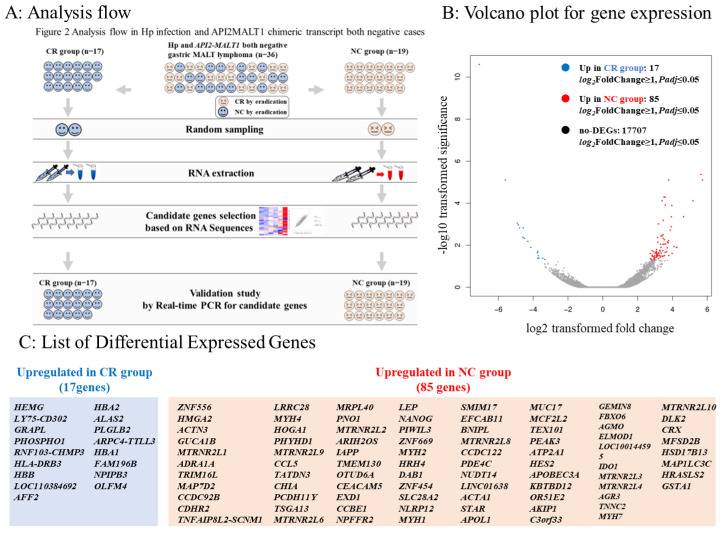
Analysis flow and differentially expressed genes in RNA sequences. (**A**): Among the 164 cases, 36 were negative for Hp infection and *API2-MALT1* mutation. We classified them into 17 cases in which CR was achieved by eradication therapy (complete response [CR] group) and 19 cases in which CR was not achieved by eradication therapy (no-change [NC] group). Two cases were randomly selected from each of the CR and NC groups. We extracted RNA from tumor sites of the selected cases and conducted a comprehensive analysis via RNA sequencing to identify genes highly expressed in the CR group and NC group. We selected candidate genes that could be predictors of eradication therapy efficacy and characterized the CR and NC groups. To determine the predictive marker genes among these candidate genes, we validated the expression of these genes using real-time PCR in 17 patients in the CR group and 19 patients in the NC group. (**B**): Volcano plot for gene expression. The X-axis represents log_2_ transformed fold change, and the Y-axis represents -log_10_ transformed significance. Red points represent upregulated DEGs. Blue points represent downregulated DEGs. Gray points represent non-DEGs. (**C**): List of differentially expressed genes (DEGs).

**Figure 3 cancers-15-01206-f003:**
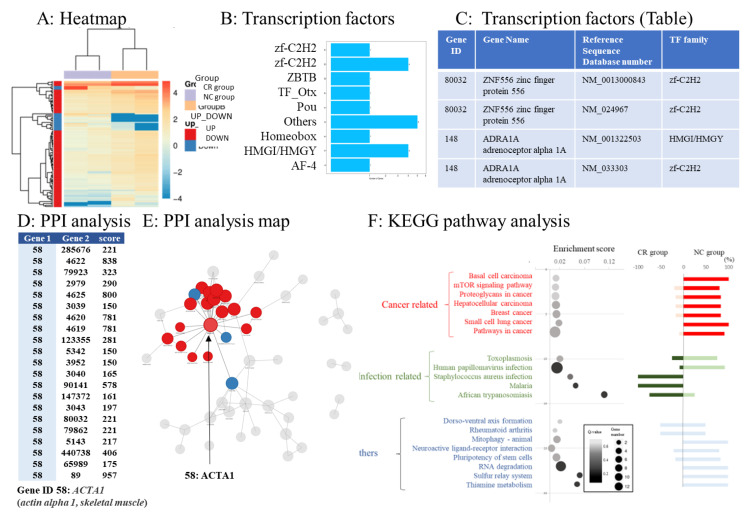
Bioinformatics analysis. (**A**) The gradient legend at the top right of the graph represents the FPKM value that has been logarithmically converted. Each column represents a sample, each row represents a gene, and different colors represent different expression levels: red for high expression and blue for low expression. The X-axis represents the comparison sample, while the Y-axis represents differentially expressed genes (DEGs). Coloring indicates log_2_ transformed fold change (red: high in the NC group, blue: high in the CR group). (**B**,**C**) List of transcription factors coded by DEGs. (**D**) Protein–protein interaction (PPI) analysis. According to the PPI analysis, actin alpha 1 and skeletal muscle (ACTA1) had the highest interaction score. The table shows the interaction score between ACTA1 protein and proteins coded by other genes according to the STRING database. Gene 1: interaction gene 1. Gene 2: interaction gene 2. The larger the score, the more reliable the result. (**E**) Visualized protein–protein interactions and enlarged view focusing on genes with high interaction scores. (**F**) Pathway functional enrichment of differentially expressed genes (DEGs) according to Kyoto Encyclopedia of Genes and Genomes (KEGG) pathway analysis. In the bubble chart, the X-axis represents the enrichment factor, while the Y-axis represents the pathway names. Color gradation of the q-value (high: white; low: black). A lower q-value indicates more significant enrichment. The point size indicates the DEG number (larger dots refer to larger amounts). The enrichment score refers to the value of the enrichment factor, which is the quotient of the foreground value (the number of DEGs) and background value (total gene amount). The larger the value, the more significant the enrichment. The bar chart on the right side shows the balance between upregulated and downregulated genes in the corresponding pathway. The top 20 pathways that fluctuated the most were sequenced according to the functional classification. The top 20 pathways were classified into cancer-related genes, infection-related genes, and others.

**Figure 4 cancers-15-01206-f004:**
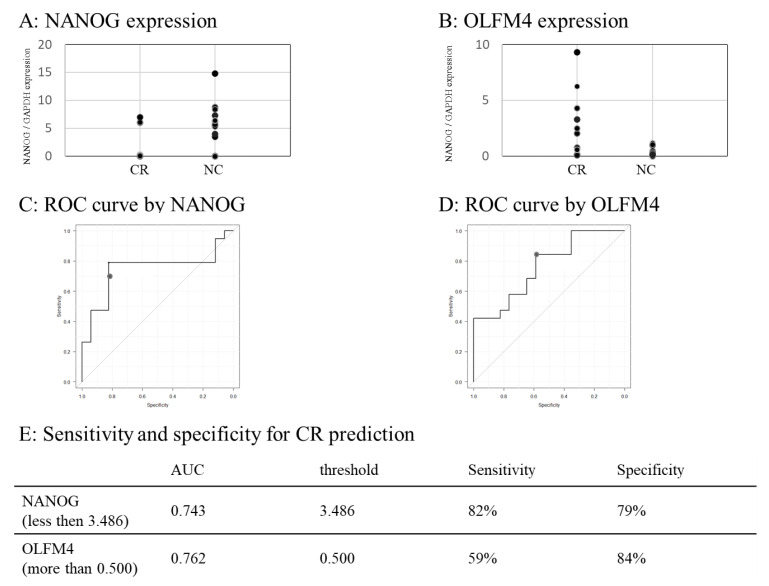
Predictability of NANOG and OLFM4 for CR. (**A**,**B**) Dot chart showing the NANOG (**A**) and OLFM4 (**B**) expression in the CR and NC groups. (**C**,**D**) ROC curves generated from NANOG expression value to predict CR (**C**) and OLFM4 (**D**). (**E**) Predictability of NANOG (less than threshold determined with the Youden index) and OLFM4 (more than threshold determined with the Youden index) to predict CR. ROC, receiver operating characteristic.

**Table 1 cancers-15-01206-t001:** Comparison between CR and NC groups in gastric MALT lymphoma negative for both Hp and *API2-MALT1* chimeric transcript.

		CR Group (*n* = 17)	NC Group (*n* = 19)	*p*-Value
Median age (range)		56.4 ± 14.0 (32–79)	64.9 ± 12.4 (39–88)	0.063
Sex	Male	6 (35%)	7 (37%)	1
	Female	11 (65%)	12 (63%)	
Mucosal atrophy	Closed-type	14 (82%)	17 (89%)	0.650
	Open-type	3 (18%)	2 (11%)	
Number of lesions	Single	5 (29%)	3 (16%)	0.434
	Multiple	12 (71%)	16 (84%)	
Location	U	2 (12%)	3 (16%)	0.727
	M/L	15 (88%)	16 (84%)	
Morphological type	Superficial	15 (88%)	17 (90%)	0.906 †
	Elevated	0	1 (5%)	0.337 †
	Other	2 (12%)	1 (5%)	0.481 †

Fisher’s exact test was performed for categorical variables. Mann–Whitney *U* test was performed for comparative analyses of continuous variables. † The significance level (*p* < 0.05/3) was adjusted with Bonferroni correction for multiple comparisons. U, upper part of stomach; M, middle part of stomach; L, lower part of stomach; CR, complete response; NC, no change.

**Table 2 cancers-15-01206-t002:** Detected candidate genes.

	Detection Method		Gene Symbol	Gene Expression (CR Group)	Gene Expression (NC Group)	Log_2_ Fold Change	*p*-Value	FDR
Upregulated in CR group	Infection-related genes	Based on KEGG pathway analysis	*HBA1*	74,654.784	2580.948	−4.854	1.08 × 10^−6^ *	0.00106 *
			*HBA2*	113,119.690	3758.04	−4.911	7.96 × 10^−7^ *	0.000886 *
			*HBB*	58,139.590	2118.052	−4.778	1.76 × 10^−6^ *	0.00149 *
			*HLA-DRB3*	1826.461	139.764	−3.707	0.000142 *	0.0313 *
		Based on DEG	*NPIPB3*	770.699	5.639	−7.094	1.40 × 10^−15^ *	2.50 × 10^−11^ *
			*ALAS2*	755.955	31.467	−4.586	6.15 × 10^−6^ *	0.00421 *
			*OLFM4*	947.359	41.761	−4.503	7.62 × 10^−^^6^ *	0.00468 *
Upregulated in NC group	Cancer-related genes	Based on KEGG pathway analysis	*TMEM130*	71.323	561.849	2.977	0.00014 *	0.0313 *
			*GSTA1*	50.275	600.022	3.577	7.08 × 10^−5^ *	0.0221 *
		Based on DEG	*LEP*	25.913	404.156	3.963	8.01 × 10^−8^ *	0.00013 *
			*MTRNR2L8*	25,678.176	270,967.1	3.399	3.87 × 10^−8^ *	7.65 × 10^−5^ *
			*NANOG*	55.876	814.437	3.865	2.37 × 10^−6^ *	0.00191 *
			*OTUD6A*	64.137	977.690	3.930	1.41 × 10^−6^ *	0.00131 *
			*ZNF669*	332.030	5072	3.933	3.49 × 10^−7^ *	0.000444 *
	Transcription factor		*ZNF556*	29.540	342.919	3.537	9.26 × 10^−6^ *	0.00549 *
	Protein–protein interaction		*ACTA1*	7.572	275.414	5.184	3.75 × 10^−8^ *	7.65 × 10^−5^ *

* Statistically significant. FDR, false discovery rate.

**Table 3 cancers-15-01206-t003:** Detected candidate genes: results of validation study based on real-time PCR.

	Detection Method		Gene Symbol	Gene Expression (CR Group)	Gene Expression (NC Group)	*p*-Value	FDR
Upregulated in CR group	Infection-related genes	Based on KEGG pathway analysis	*HBA1*	1.03	0.52	0.343	0.686
			*HBA2*	1.47	0.40	0.236	0.629
			*HBB*	2.91	1.66	0.564	0.752
			*HLA-DRB3*	0.00	0.01	0.6	0.738
		Based on DEG	*NPIPB3*	0.11	0.04	0.033 *	0.132
			*ALAS2*	0.06	0.00	0.112	0.358
			*OLFM4*	1.84	0.32	0.005 *	0.04 *
Upregulated in NC group	Cancer-related genes	Based on KEGG pathway analysis	*TMEM130*	0.82	0.72	0.887	0.887
			*GSTA1*	1.95	2.57	0.785	0.837
		Based on DEG	*LEP*	0.23	0.28	0.382	0.611
			*MTRNR2L8*	0.67	0.60	0.768	0.878
			*NANOG*	1.36	6.11	0.003 *	0.048 *
			*OTUD6A*	0.20	0.27	0.286	0.654
			*ZNF669*	0.28	0.98	0.365	0.649
	Transcription factor		*ZNF556*	0.26	0.52	0.018 *	0.096
	Protein–protein interaction		*ACTA1*	0.25	0.67	0.383	0.557

* Statistically significant. FDR, false discovery rate.

## Data Availability

The data supporting the findings of this study are available from the corresponding author upon reasonable request.

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
