# Peer review of "Comprehensive Analysis of Gene Expression Profiling to Explore Predictive Markers for Eradication Therapy Efficacy against Helicobacter pylori-Negative Gastric MALT Lymphoma"

_cancers, 2023, doi:10.3390/cancers15041206_

Round 1

Reviewer 1 Report

The authors provide very interesting and original data regarding CR prediction based on genomics for localized MALT gastric lymphoma. The manuscript is well written and its limitations (retrospective data, preliminary study, low sample size for RNA-seq, no functional experiments to confirm impact of genetics findings/aberrations) are discussed. However, the study lacks from NGS data to assess the mutational spectrum of their patients and to look for eventual correlations between GEP profile and somatic variants.

The authors should better discuss the routine applicability of their findinds for hematologists/clinicians. Do they anticipate their findings may be considered as a companion diagnostic tool for next patients with MALT gastric lymphoma, before starting treatment ?

Author Response

[Response]

We are grateful for the positive review of our manuscript and for the constructive comments and useful suggestions.

We have revised the manuscript in response to the reviewers' comments; point-by-point responses to your comments.

Our responses are written in red font. Revisions of the manuscript have been conducted in tracking mode and indicated in red font.

The authors provide very interesting and original data regarding CR prediction based on genomics for localized MALT gastric lymphoma. The manuscript is well written and its limitations (retrospective data, preliminary study, low sample size for RNA-seq, no functional experiments to confirm impact of genetics findings/aberrations) are discussed. However, the study lacks from NGS data to assess the mutational spectrum of their patients and to look for eventual correlations between GEP profile and somatic variants.

[Response]

Thank you very much for your invaluable comments. As you pointed out, we agree that the lack of NGS analysis on mutations is a major limitation of this study.

It is certain that the integration of findings of the GEP profile obtained in this study with the mutation status by NGS could have provided more precise prediction accuracy and made this study more useful. We have added this point to the discussion section (lines 443-447).

The authors should better discuss the routine applicability of their findinds for hematologists/clinicians. Do they anticipate their findings may be considered as a companion diagnostic tool for next patients with MALT gastric lymphoma, before starting treatment ?

[Response]

Thank you for bringing this important point into discussion. As you pointed out, we should have mentioned and highlighted the routine applicability of our findings for hematologists/clinicians more carefully. We believe that our study could give insights for future studies to clarify the pathogen of Hp-negative gastric MALT lymphoma and help identify more accurate predictive markers in the future, but the sensitivity and specificity of our proposed markers are not high enough for clinical application as it is. As mentioned in the discussion section (lines 430-434), some validation studies with immunohistochemistry or some other procedures will be vital as a step before clinical application. We have included these points in the discussion section (lines 426―430).

Reviewer 2 Report

Overall I think this work is very interesting, and highlighting differential expression of NANOG and OLFM4 perhaps sheds some light into the pathogenesis of H. Pylori negative gastric MALT.

Introduction:

The first two sentences are awkwardly worded because most gastric MALT are Hp+ but only about 20-25% have t(11;18).  Therefore, I would change the second sentence to say XXX% of gastric MALT are Hp negative and XXX% do not harbor the API2-MALT translocation.

2. Materials and Methods

2.1 Patients:

  - Please revise this selection to mention selection criteria (for instance, patients with gastric MALT lymphoma at our center between 10/2006-9/2020 and who received eradication therapy as frontline treatment were included.  Patients with other histologies (DLBCL, FL) and with non-gastric MALt were excludes as were patients who received any other frontline treatment.).

  - Please move the # of patients, # of excluded patients, etc. to the Results section. 

2.2 Evaluation...

Please move the information regarding # of patients (137 patients, table 1) to Results.

3. Results

3.1 Change section title to Patients

Add here "164 patients were identified...14 excluded due to XXX reasons, etc."

3.2. Add "137 evaluable patients..." from 2.2 above here and table 1 should be moved here.

In the discussion, I would explicitly mention that the predictive values for NANOG and OLFM4 are not strong enough for clinical use and that the data are hypothesis-generating only.

I would also consider looking at NGS as the spectrum of mutations in gastric MALT has been previously defined; perhaps this would make the prediction model more robust.  This analysis obviously does not need to be run for this paper, but it could be considered as part of future analyses.

Also, the supplementary methods mention that two courses of eradication were done in some Hp+ patients (only for persistent Hp positivity) -- this could be a confounder.  I would mention in the methods section of the paper ("Hp+ cases with persistent Hp after frontline eradication therapy were eligible for second-line eradication therapy.") and, if possible, in the results section would mention how many Hp+ patients had second-line therapy.

Table S1: add the word pylori to Helicobacter to the name of the table since this is specifically Hp.  Add abbreviations to the legend (Hp, CR).  Add a hyphen between API2 and MALT (API2-MALT instead of API2MALT)

Author Response

[Response]

We are grateful for the positive review of our manuscript and for the constructive comments and useful suggestions.

We have revised the manuscript in response to the reviewers' comments; point-by-point responses to your comments.

Our responses are written in red font. Revisions of the manuscript have been conducted in tracking mode and indicated in red font.

Overall I think this work is very interesting, and highlighting differential expression of NANOG and OLFM4 perhaps sheds some light into the pathogenesis of H. Pylori negative gastric MALT.

 Introduction:

The first two sentences are awkwardly worded because most gastric MALT are Hp+ but only about 20-25% have t(11;18).  Therefore, I would change the second sentence to say XXX% of gastric MALT are Hp negative and XXX% do not harbor the API2-MALT translocation.

[Response]

Thank you very much for your recommendation. According to your suggestion, I have revised the sentences as follows.

“However, 23% of gastric MALT lymphoma are Hp negative and 93% do not harbor the API2-MALT1 translocation in stage I gastric MALT lymphoma [2]”. (lines 49-51)

  1. Materials and Methods

2.1 Patients:

  - Please revise this selection to mention selection criteria (for instance, patients with gastric MALT lymphoma at our center between 10/2006-9/2020 and who received eradication therapy as frontline treatment were included.  Patients with other histologies (DLBCL, FL) and with non-gastric MALt were excludes as were patients who received any other frontline treatment.).

[Response]

Thank you for your invaluable advice. According to your recommendation, I have revised the first sentence of the Methods 2.1 Patients section as follows:

“Patients with gastric MALT lymphoma treated at our center between October 2006 and September 2020 and who received eradication therapy as frontline treatment were included. Patients who received any other frontline treatment and lost follow-up cases were excluded.” (lines 94-97)

  - Please move the # of patients, # of excluded patients, etc. to the Results section. 

[Response]

Thank you for your in-detailed suggestion. I have moved the sentences you pointed out to the Results section 3.1 (lines 185-189). Along with this move, I have moved the position of Figure 1.

2.2 Evaluation...

Please move the information regarding # of patients (137 patients, table 1) to Results.

[Response]

I have moved the description about 137 patients and Table 1 to the Result section 3.1 (lines 185-189) and moved table 1 below the Result section 3.3. Thank you.

  1. Results

3.1 Change section title to Patients

Add here "164 patients were identified...14 excluded due to XXX reasons, etc."

[Response]

I have changed the title of section 3.1 to “Patients” and added the sentence there as you instructed (lines 185-189). Thank you.

3.2. Add "137 evaluable patients..." from 2.2 above here and table 1 should be moved here.

[Response]

Thank you for your helpful recommendation, We have added the description “137 evaluable patients…” to 3.2 (line 216). The detailed backgrounds and treatment outcomes of 137 patients are described in Table S1. Thus, we have added the caption for Table S1 in that section (line 223). We have also moved Table 1 below section 3.3 along with the caption in the main text (line 232).

In the discussion, I would explicitly mention that the predictive values for NANOG and OLFM4 are not strong enough for clinical use and that the data are hypothesis-generating only.

 [Response]

Thank you for bringing this important point into discussion. As you pointed out, our data generated only the hypothesis and is insufficient to show the usefulness of these predictive markers in clinical practice from the viewpoint of diagnostic accuracy and lack of validation. We have added several sentences explaining these points in the discussion section (lines 426-430).

I would also consider looking at NGS as the spectrum of mutations in gastric MALT has been previously defined; perhaps this would make the prediction model more robust.  This analysis obviously does not need to be run for this paper, but it could be considered as part of future analyses.

 [Response]

Thank you for pointing out a very important issue. As you pointed out, the lack of NGS analysis on mutations is a major limitation of this study. We completely agree with you and believe that the integration of findings of the gene expression profiles obtained in this study with the mutation status by NGS analysis could provide more precise prediction accuracy and could have made this study more useful. These points should be mentioned as limitations and proposed as an issue for the future. We have added these points to the discussion (lines 443-447).

Also, the supplementary methods mention that two courses of eradication were done in some Hp+ patients (only for persistent Hp positivity) -- this could be a confounder.  I would mention in the methods section of the paper ("Hp+ cases with persistent Hp after frontline eradication therapy were eligible for second-line eradication therapy.") and, if possible, in the results section would mention how many Hp+ patients had second-line therapy.

 [Response]

Thank you for your helpful recommendation. As you pointed out, the difference in the number of eradication therapy could be a potential bias for this study. Thus, we mentioned it in methods section 2.3 of the main manuscript briefly (lines 113-117). In result section 3.1, we also described the number of Hp-positive patients who underwent second-line eradication therapy and therapy outcomes for Hp infection in Hp-positive cases according to your instructions (lines 193-196).

Table S1: add the word pylori to Helicobacter to the name of the table since this is specifically Hp.  Add abbreviations to the legend (Hp, CR).  Add a hyphen between API2 and MALT (API2-MALT instead of API2MALT)

 [Response]

We apologize for our oversight and thank you for your comments. I have changed the title of Table S1 as you instructed. And I have added the abbreviation to the legend of Table S1 and put a hyphen in between API2 and MALT1. Thank you.